# Noninvasive Assessment of Impaired Gas Exchange with the Alveolar Gas Monitor Predicts Clinical Deterioration in COVID-19 Patients

**DOI:** 10.3390/jcm12196203

**Published:** 2023-09-26

**Authors:** W. Cameron McGuire, Alex K. Pearce, Ann R. Elliott, Janelle M. Fine, John B. West, Daniel R. Crouch, G. Kim Prisk, Atul Malhotra

**Affiliations:** UC San Diego Health Division of Pulmonary, Critical Care, Sleep Medicine, and Physiology 9500 Gilman Drive, Mail Code 7381, La Jolla, CA 92093, USA; apearce@health.ucsd.edu (A.K.P.); aelliott@health.ucsd.edu (A.R.E.); jfine@health.ucsd.edu (J.M.F.); jwest@health.ucsd.edu (J.B.W.); dcrouch@health.ucsd.edu (D.R.C.); kprisk@health.ucsd.edu (G.K.P.); amalhotra@health.ucsd.edu (A.M.)

**Keywords:** applied physiology, COVID-19, gas exchange, hypoxia, respiratory failure, supplemental oxygen, alveolar gas monitor

## Abstract

Background and Objective: The COVID-19 pandemic magnified the importance of gas exchange abnormalities in early respiratory failure. Pulse oximetry (SpO_2_) has not been universally effective for clinical decision-making, possibly because of limitations. The alveolar gas monitor (AGM100) adds exhaled gas tensions to SpO_2_ to calculate the oxygen deficit (OD). The OD parallels the alveolar-to-arterial oxygen difference (AaDO_2_) in outpatients with cardiopulmonary disease. We hypothesized that the OD would discriminate between COVID-19 patients who require hospital admission and those who are discharged home, as well as predict need for supplemental oxygen during the index hospitalization. Methods: Patients presenting with dyspnea and COVID-19 were enrolled with informed consent and had OD measured using the AGM100. The OD was then compared between admitted and discharged patients and between patients who required supplemental oxygen and those who did not. The OD was also compared to SpO_2_ for each of these outcomes using receiver operating characteristic (ROC) curves. Results: Thirty patients were COVID-19 positive and had complete AGM100 data. The mean OD was significantly (*p* = 0.025) higher among those admitted 50.0 ± 20.6 (mean ± SD) vs. discharged 27.0 ± 14.3 (mean ± SD). The OD was also significantly (*p* < 0.0001) higher among those requiring supplemental oxygen 60.1 ± 12.9 (mean ± SD) vs. those remaining on room air 25.2 ± 11.9 (mean ± SD). ROC curves for the OD demonstrated very good and excellent sensitivity for predicting hospital admission and supplemental oxygen administration, respectively. The OD performed better than an SpO_2_ threshold of <94%. Conclusions: The AGM100 is a novel, noninvasive way of measuring impaired gas exchange for clinically important endpoints in COVID-19.

## 1. Introduction

The COVID-19 pandemic has had devastating consequences for patients and families around the world. Over one million people have died in the US alone, with countless others around the world [1]. The pandemic overwhelmed the healthcare system, challenging the supply of equipment and personnel for the care of the large volume of patients [2]. This situation highlighted the limited capacity of the healthcare system and emphasized the need for more efficient triage should another global pandemic or surge occur [3].

The COVID-19 pandemic was particularly challenging since some patients were not aware of their severity of illness. Patients with so-called “happy hypoxia” presented with marked hypoxemia to the surprise of both patients and providers [4]. Thus, pulse oximetry use became widespread during the pandemic, with many patients self-monitoring using this technology [5]. However, pulse oximetry has limitations based on skin pigmentation, presence of peripheral vascular disease, poor perfusion, metabolic derangements, and inherent inaccuracies as compared to arterial blood gas (ABG) measurements [6,7,8]. Thus, novel, noninvasive strategies to assess gas exchange would be desirable.

The alveolar gas monitor (AGM100) is a new technology (MediPines, Yorba Linda, CA, USA) that was developed to assess pulmonary gas exchange efficiency and obviate the need for ABG measurements in some cases [9,10]. It has received Food and Drug Administration approval for this purpose [11] and is being used by an increasing number of investigators for a variety of clinical and research applications [12,13]. The AGM100 has been validated in healthy controls and in patients with stable cardiopulmonary disease and quantifies the oxygen deficit (OD), which is a surrogate for the alveolar arterial oxygen difference (A-aDO_2_) colloquially known as the A-a gradient. Prior studies have used the AGM100 for a variety of cardiopulmonary conditions [14,15] and a variety of environmental extremes [16,17,18], but its role in acute illnesses, specifically COVID-19, is less clear.

The OD is a noninvasive assessment that uses end-tidal carbon dioxide tension, end-tidal oxygen tension, and pulse oximetry (SpO_2_) to estimate the A-aDO_2_. We aimed to determine if the OD would help to predict the risk of clinical deterioration as defined by need for hospital admission and need for supplemental oxygen in patients with COVID-19.

We hypothesized that patients with large oxygen deficits would be at high risk of progressive clinical deterioration as compared to patients with small oxygen deficits. We further surmised that the OD would perform at least as effectively as SpO_2_ (the current standard of care) in triaging patients with COVID-19 at risk for clinical deterioration. If these hypotheses were correct, the AGM100 and its output, the OD, may be a useful point-of-care tool to identify patients who require immediate healthcare interventions as compared to those who could be managed expectantly.

## 2. Methods

### 2.1. Study Design

A prospective diagnostic accuracy study was performed comparing an index test (OD) to a reference standard (SpO_2_) in predicting clinical deterioration as defined by the need for hospital admission and need for supplemental oxygen. This study was approved by our university’s Institutional Review Board, and all patients provided signed, informed consent.

### 2.2. Participants

Patients 18 years or older with documented or suspected COVID-19 pneumonia in the emergency department at two different academic medical centers were approached for the study between November 2020 and January 2022. Potential participants were identified by review of the emergency room tracking board for a chief complaint of dyspnea. Those on room air or low-dose supplemental oxygen (defined as 4 L per minute or less by nasal cannula) were eligible for enrolment. All potential participants had to have a positive COVID-19 test or a high pre-test probability for COVID-19 based on symptoms with a confirmatory test pending. Medication records were reviewed to ensure patients were not taking any medications that may interfere with pulse oximetry readings (e.g., vasodilators). Convenience sampling was performed as many patients were on too much supplemental oxygen by the time of potential study enrolment and because the primary investigator who collected all data was clinically active in the intensive care unit during the study enrolment period.

### 2.3. Test Methods

The index test was the oxygen deficit (OD) as calculated by the AGM100. We have previously described the derivation of the OD [10]. In brief, it is derived by subtracting the calculated arterial PO_2_ (gPaO_2_) from the measured alveolar PO_2_ (PAO_2_), which is obtained from expired breath using rapid gas analyzers integrated into the AMG100. We have also previously described the calculation of gPaO_2_ by the AGM100 [9]. In brief, it is a logarithm of the Hill equation [9], which has previously been used by Severinghaus [19,20,21]. The determination of the OD also accounts for the effect of arterial CO_2_ on pH and thereby on the oxygen hemoglobin dissociation curve. This is achieved by use of Kelman subroutines [22,23] to correct the arterial CO_2_ for the alveolar CO_2_ operating under Comroe’s assumption that alveolar CO_2_ is equivalent to arterial CO_2_ [24]. A limitation of the AGM100 is the lack of accuracy of the OD and the gPaO_2_ when the SpO_2_ is high, so patients with an SpO_2_ of 100% were excluded from enrollment or analysis. The OD and gPaO_2_ were considered exploratory variables of interest, meaning no prespecified cut-off values were established as these variables are unique to the AGM100, and this study was a novel application of this technology without a known relationship to the outcomes of interest.

The reference test was SpO_2_, which is standard of care in all hospitals across the country. A pre-specified threshold of SpO_2_ < 94% was used to compare efficacy of the OD in predicting clinically relevant outcomes such as hospital admission or need for supplemental oxygen. This SpO_2_ threshold represents the midpoint of the range recommended by the National Institutes of Health when caring for COVID-19 patients [25]. Additionally, 94% is well within a safe range that avoids the harms of hypoxia and hyperoxia [26,27]. Finally, 94% is sufficiently high enough to avoid occult hypoxemia in most individuals [7], even when accounting for the absolute mean error and standard deviation of most pulse oximeters [28,29]. While pulse oximetry is the standard of care for screening for hypoxia, clinical decisions about the outcomes of interest (hospital admission and supplemental oxygen provision) are likely more complex than a singular SpO_2_ value. For this reason, we felt a prespecified value in the midrange was more faithful to the heterogeneity of clinical decision making than an artificially low or high value for SpO_2_.

For the study patients who had already been placed on low-dose oxygen by their clinical care team, supplemental oxygen was temporarily removed with Institutional Review Board (IRB) approval so that AGM100 measurements could be obtained on room air and compared to SpO_2_ under the same conditions. Measurements were made after five minutes of tidal breathing on room air to ensure no residual increased alveolar oxygen tension existed from prior supplemental oxygen. The pulse oximeter integrated into the AGM100 was placed on a digit of the same extremity as the hospital-approved pulse oximeter used for standard clinical care.

Throughout the study, the hospital pulse oximeter was left on the patient to monitor for hypoxia as part of the study safety protocol. Any patient who desaturated to an SpO_2_ < 88% on room air on the hospital pulse oximeter was excluded from the study, and if supplemental oxygen was removed as a part of the study, it was immediately replaced. All measurements were made with the patient sitting upright in bed at 45 degrees to standardize functional residual capacity and improve gas exchange as compared to the supine position. All decisions about hospital admission or discharge and provision of supplemental oxygen were made by the clinical care team without any knowledge of the AGM100 readings and without any input from the primary investigator (WCM). Patients who had previously been administered supplemental oxygen were only considered to have met the supplemental oxygen outcome if oxygen was administered again during the index hospitalization after the AGM100 measurements were obtained. Patients were only considered to have met the hospital admission outcome if they were admitted to the hospital for at least 24 h after the AGM100 measurement. Hospital admission was determined based on usual practice and clinical decision-making of the admitting teams. Clinical care teams were blinded to the performance and nature of the experimental study.

### 2.4. Statistical Analysis

Descriptive data were analyzed using Microsoft Excel (Version 16.60, 2022) with an unpaired, two-tailed *t*-test with equal variance. Receiver operating characteristic (ROC) curves were generated for gPaO_2_ as a continuous variable without a prespecified threshold, OD as a continuous variable without a prespecified threshold, and SpO_2_ as a continuous variable with a prespecified threshold of 94%, as previously mentioned, using SPSS (Version 28.0.1.1, 2021, IBM Corporation, Armonk, NY, USA).

Because an increasing OD is representative of worsening gas exchange while an increasing gPaO_2_ and SpO_2_ are representative of improved gas exchange, a dummy variable, OD flip, was created by subtracting the OD from 100. This allowed the ROC curves for all three predictors (SpO_2_, gPaO_2_, and OD flip) to be plotted on the same axis for visual clarity and comparison.

Due to the correlated nature of the variables, the DeLong method [30] was applied to the area under the ROC curves for all three classifiers using R (R Core Team, 2022) to assess for a statistically significant difference between the ROC for SpO_2_ and either of the AGM100 variables of interest.

## 3. Results

We enrolled 39 subjects, of whom 8 ultimately did not have COVID-19 when their confirmatory testing returned. Three of the eight patients also had an SpO_2_ of 100%, so their data were excluded. Of the COVID-19-positive (n = 31) patients, 30 were included in the final analysis. One COVID-19-positive patient was excluded because of incomplete AGM100 data. Patient characteristics (n = 30) can be seen in Table 1. Aside from diabetes, hypertension, and heart failure, no other comorbidities occurred at a frequency greater than 10 percent of the study population. In all 30 patients, the decision about hospital admission or supplemental oxygen provision was made within 24 h of AGM100 measurements.

Among those patients admitted to the hospital (n = 25), the mean OD was 50.0 ± 20.6 (mean ± SD), while among those discharged from the ED (n = 5), the mean OD was 27.0 ± 14.3 (mean ± SD), a difference that was statistically significant (*p* = 0.025) (See Figure 1).

Analysis of ROC curves for hospital admission demonstrated an area under the curve (AUC) of 0.832 (±0.085), which was statistically significant (*p* < 0.001; 95%CI 0.665, 0.999). The overall model quality was 0.67, and an OD > 31 predicted the need for hospital admission with a sensitivity of 0.8 and a 1 - specificity of 0.8 (See Figure 2). The OD had a higher AUC than the SpO_2_ in predicting the need for hospital admission though the confidence intervals of both AUCs overlapped (See Figure 3).

Among those patients requiring supplemental oxygen (n = 18), the OD was 60.1 ± 12.9 (mean ± SD), while among those not requiring supplemental oxygen (n = 12), the OD was 25.2 ± 11.9 (mean ± SD), which was statistically significant (*p* < 0.0001) (See Figure 1). Analysis of ROC curves for supplemental oxygen administration demonstrated an AUC of 0.981 (±0.021), which was statistically significant (*p* < 0.001; 95% CI 0.949, 1.023). The overall model quality was 0.94, and an OD > 37 predicted the need for supplemental oxygen administration with a sensitivity of 0.917 and a 1 - specificity of 1.0 (See Figure 4). The OD had a higher AUC than the SpO_2_ in predicting the need for supplemental oxygen though the confidence intervals of both AUCs overlapped (See Figure 5). A DeLong test between the AUC for OD flip and SpO_2_ did not reach statistical significance (Z = 1.36; *p* = 0.17; 95% CI −0.03, 0.19).

There were no adverse events for any participants in the study and all individuals maintained an SpO_2_ ≥ 88% for the duration of AGM100 recordings performed on room air. One individual progressed to needing high-flow nasal oxygen during the index hospitalization. No one required noninvasive or invasive ventilation, extracorporeal membrane oxygen, or ICU level of care.

## 4. Discussion

The A-a gradient is a useful concept in respiratory physiology. It can distinguish various causes of impaired gas exchange since the gradient is elevated with ventilation/perfusion mismatch or with shunt. In contrast, the value is typically normal in patients with hypoventilation. The oxygen deficit (OD) has previously been studied as a surrogate for the A-a gradient in stable outpatients with various cardiopulmonary conditions [15]. In this study, we examined the novel application of this technology in acutely ill ED patients with COVID-19.

Our findings are important for several reasons. We have observed an excellent AUC for the analysis of AGM100 data regarding the risk of respiratory deterioration in COVID-19-positive patients. This finding suggests that the AGM100 may be a useful method to triage patients based on risk of respiratory decompensation. The data demonstrate the feasibility of using the AGM100 in hospitalized patients and suggest that more widespread study of the technique may be valuable for risk stratification in general. Moreover, the OD performed better than an SpO_2_ of 94% in ROC analysis at predicting the need for supplemental oxygen administration. It may, therefore, represent a new, noninvasive diagnostic strategy in early respiratory disease, and we speculate that the addition of exhaled gas tensions to the risk stratification picture may be a novel way to address some of the limitations in pulse oximetry to detect occult hypoxemia.

Despite our study’s strengths, we acknowledge several limitations. First, our sample size was modest, from two institutions, and thus, we are supportive of more widespread efforts to extend or to refute our findings. Nonetheless, we studied a heterogeneous group of individuals in a real-world setting, and thus, we anticipate that our findings will be generalizable to other cohorts. Second, we used as our outcome measure the need for escalation of therapy based on supplemental oxygen administration during hospitalization. One could argue that poor gas exchange is predictive of the need for supplemental oxygen, and the AGM100 is not required to make this assertion. However, the AGM100 has value since it performed better than pulse oximetry with AUC curves showing a trend toward superiority despite overlapping confidence intervals and a nonsignificant result using the DeLong method. Since SpO_2_ is typically relied upon in the context of clinical decision-making, the OD has the potential to provide a more accurate assessment if studied prospectively in a larger sample size. Moreover, the very high predictive value that we observed provides some reassurance that decisions could be made based on this technology. Third, our outcome of need for supplemental oxygen during the hospitalization was not temporally linked to the measurement of the OD. If supplemental oxygen was ultimately administered long after the OD was measured, other contravening factors may have played a role in the change in the patient’s clinical status. Nonetheless, the ability to predict the need for supplemental oxygen administration is an important outcome worth further investigation. Fourth, we recognize the dynamic nature of the COVID pandemic and the changing behavior of the viral variants in heterogeneous populations with highly variable immune protection (vaccinated, boosted, prior infection, etc.). Thus, our technique would need to be studied in a context-specific manner and for other infections. Nevertheless, the concepts are useful and could be applied to future surges or subsequent respiratory epidemics.

We have also identified several future directions to take our research. First, we believe that the addition of exhaled gas tensions to SpO_2_ may be a way to address the race-based limitations of pulse oximetry and believe this should be studied prospectively. Although the AGM100 relies on a pulse oximeter, this would create bias toward the null hypothesis that the OD is no better than SpO_2_ alone. If, however, there is a signal in favor of the OD relative to SpO_2_ in patients at risk for occult hypoxemia, this finding would suggest the added value of expired gas tensions and provide a rapid, noninvasive way to address this important issue. Second, there may be a dose response between the OD and the fraction of expired oxygen (FiO_2_) administered to hypoxemic patients. Current clinical practice relies on titrating the FiO_2_ based on improvement in the SpO_2_, but given the aforementioned limitations in SpO_2_, our findings may suggest an alternative approach by using the OD to determine how much supplemental oxygen is required. Specifically, the FiO_2_ could be calculated based on the available AaDO_2_ from the OD. Third, the OD trend over time in hospitalized patients could inform decisions about readiness for hospital discharge, need for supplemental oxygen on discharge, and FiO_2_ prescription at time of discharge.

## 5. Conclusions

The oxygen deficit (a surrogate for the alveolar-to-arterial oxygen difference), as calculated by the alveolar gas monitor, provides a useful tool for clinical triage of COVID-19-positive patients that performs at least as well as, if not better than, pulse oximetry. An oxygen deficit > 31 is modestly sensitive for needing hospital admission and an oxygen deficit > 37 is highly sensitive for needing supplemental oxygen administration during the hospitalization.

## Figures and Tables

**Figure 1 jcm-12-06203-f001:**
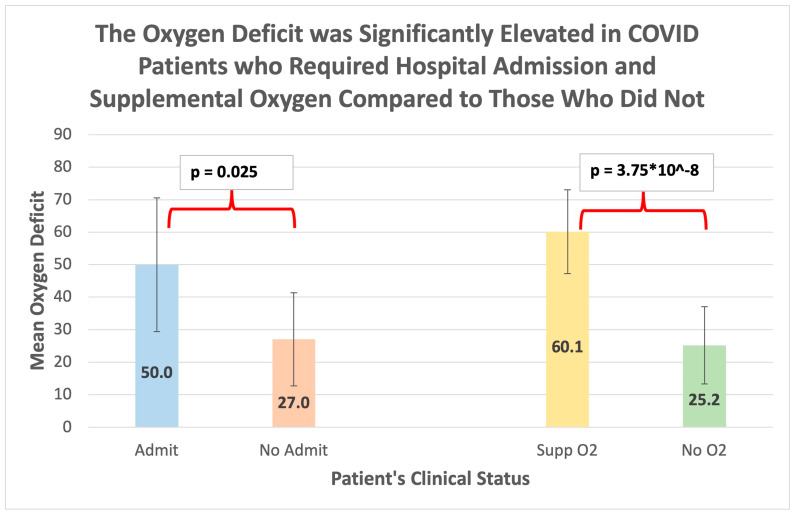
Differences in the mean oxygen deficit (OD) between those requiring hospital admission (n = 25) and those discharged from the emergency department (n = 5) and between those requiring supplemental oxygen (n = 18) and those remaining on room air (n = 12) among the COVID-19 patients included in the final analysis (n = 30). Scale bars represent the standard deviation of the mean.

**Figure 2 jcm-12-06203-f002:**
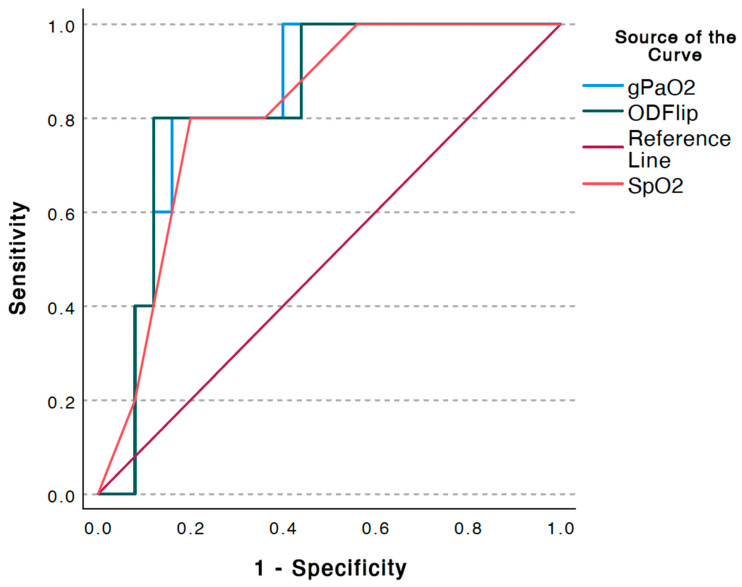
Receiver operating characteristic curves for pulse oximetry (SpO_2_), calculated arterial oxygen pressure (gPaO_2_), and oxygen deficit (OD) and their predictive ability for hospital admission. Note: OD flip is a dummy variable created as the inverse of OD so all three curves could be plotted on the same axis for visual clarity.

**Figure 3 jcm-12-06203-f003:**
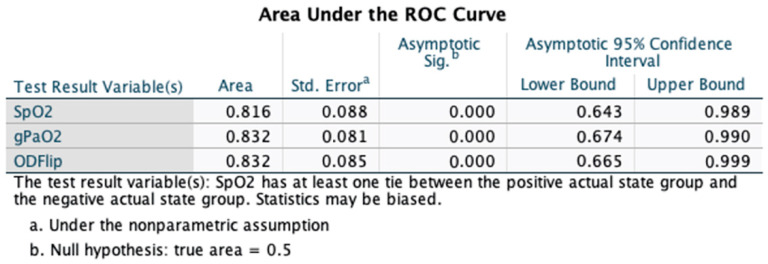
The area under the receiver operating characteristic (ROC) curve (AUC) for oxygen deficit (OD) was slightly greater than the AUC for pulse oximetry (SpO_2_) alone in predicting need for hospital admission though lacked statistical significance based on overlapping confidence intervals.

**Figure 4 jcm-12-06203-f004:**
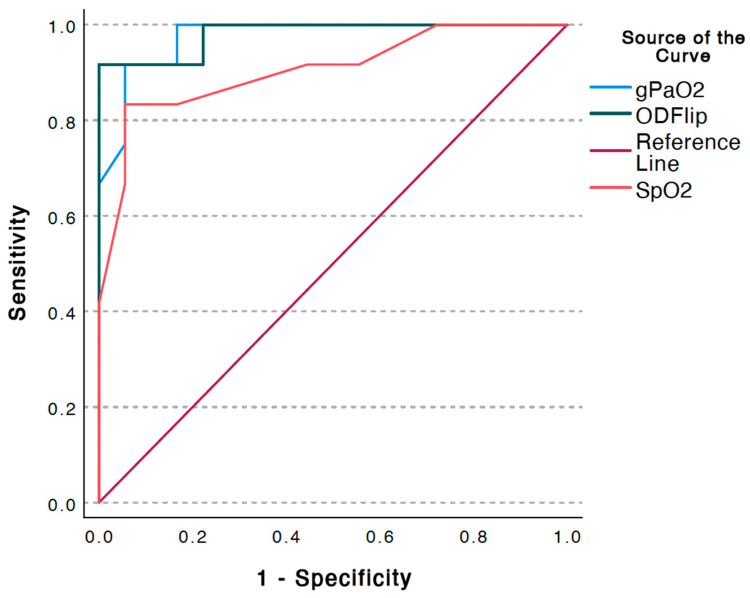
Receiver operating characteristic (ROC) Curves for pulse oximetry (SpO_2_), calculated arterial oxygen pressure (gPaO_2_), and oxygen deficit (OD) and their predictive ability for need for supplemental oxygen. Note: OD flip is a dummy variable created as the inverse of OD so all three curves could be plotted on the same axis for visual clarity and comparison.

**Figure 5 jcm-12-06203-f005:**
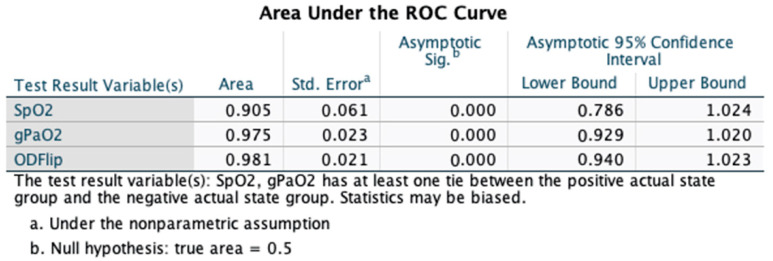
The area under the receiver operating characteristic (ROC) curve (AUC) for oxygen deficit (OD) was greater than the AUC for pulse oximetry (SpO_2_) alone in predicting need for supplemental oxygen administration and was statistically significant.

**Table 1 jcm-12-06203-t001:** Characteristics for all 30 patients included in the study. Overweight was defined as a body mass index > 24.99 kg/m^2^.

Characteristic	Value
Median Age (Range)—years	63 (34–85)
Sex—no (%)	
Male	18 (60)
Female	12 (40)
Race—no (%)	
White	23 (76.7)
Black	4 (13.3)
Asian	3 (10)
Ethnicity—no (%)	
Hispanic	7 (23.3)
Not Hispanic	23 (76.7)
Overweight—no (%)	21 (70)
Overall Mean BMI (st. dev.)—kg/m^2^	29.2 (6.72)
Diagnosis of Diabetes—no (%)	8 (26.7)
Diagnosis of Hypertension—no (%)	6 (20)
Diagnosis of Heart Failure—no (%)	5 (16.7)
Vaccination Status for COVID-19—no (%)	
Vaccinated	10 (33.3)
Not Vaccinated	20 (66.7)

## Data Availability

The data that support the findings of this study are available from the corresponding author upon reasonable request.

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
