# Peer review of "Noninvasive Assessment of Impaired Gas Exchange with the Alveolar Gas Monitor Predicts Clinical Deterioration in COVID-19 Patients"

_jcm, 2023, doi:10.3390/jcm12196203_

Round 1
Reviewer 1 Report
I have read the article by McGuire et al. with great interest. The authors evaluated oxygen deficit (OD) in patients with COVID-19.
Comments:
· Participants. Some participants were already admitted (they were recruited on the medical wards). Admission was one of the measured outcomes. Could you please clarify this discrepancy?
· Test Methods. Please, specify admission criteria. This is the primary outcome of the study.
· Statistical analysis. Please, provide a priori power calculations.
· Table 1. Only diabetes is mentioned as a comorbidity. Please, give the full list of comorbid disorders.
· Table 1. Was any of the patient taking medications (i.e. vasodilators) which could affect pulse oximetry readings?
· Table 1. Do you ABG values. Did anybody come with Type 2 respiratory failure? How would OD perform in these patients?
· Results. Figures 3 and 5. Could you please compare the AUCs of the SpO2 and OD? Consider DeLong test. There is an overlap in CIs of the AUCs, therefore I feel the curves are not different.
Author Response
I have read the article by McGuire et al. with great interest. The authors evaluated oxygen deficit (OD) in patients with COVID-19.
Comments:
- Participants. Some participants were already admitted (they were recruited on the medical wards). Admission was one of the measured outcomes. Could you please clarify this discrepancy.
We apologize for the confusion. We were stating that patients in the ED or medical ward were eligible for enrollment. The patients analyzed in this manuscript were not yet admitted to the hospital. We have clarified the wording in the manuscript by removing the confusing phrase. Additionally, the main outcome of interest was supplemental oxygen administration since the criteria used for hospital admission are highly variable (for example, a lung transplant patient was admitted for “observation” despite normal vital signs, labs, and an unconcering clinical history).
- Test Methods. Please, specify admission criteria. This is the primary outcome of the study.
The primary outcome of the study was supplemental oxygen administration not hospital admission. Hospital admission was more of an exploratory outcome as we realized during our study that US hospitals were overwhelmed by the surge of COVID, and we were hoping to identify ways to assist our emergency medicine colleagues in hospital throughput (i.e., could decisions about discharge from the ED be safely made based on the OD?). We are also unable to specify criteria for admission OR supplemental oxygen administration as these decisions were made by the treating physicians who were unaware the study was being performed to maintain blinding. We have included a statement about this in the methods section to clarify.
- Statistical analysis. Please, provide a priori power calculations.
We are not familiar with a prior study that used the AGM100 in predictive modeling. In our previous paper (https://doi.org/10.1016/j.chest.2018.02.001), the OD was significantly higher among 17 patients with pulmonary disease as compared to 31 normal subjects. Our initial approach with no a priori knowledge of the prevalence of COVID early in the study period nor knowledge of the fraction of patients who would require supplemental oxygen was to target close to twice the number of our previous paper, which is why we enrolled 39 initially.
- Table 1. Only diabetes is mentioned as a comorbidity. Please, give the full list of comorbid disorders.
We are not certain if the reviewer wants this entire list included in tabular format but have provided a complete list of the patient (n=30) comorbidities below. We have included two additional rows in our table for comorbidities that occurred in greater than 10% of patients (hypertension and heart failure) and that are also linked with worse outcomes in COVID-19 infection. We have explicitly stated that all other comorbidities (listed below) did not occur with great enough frequency to be mentioned. We are happy to discuss modifying this further but feel an exhaustive table will be unwieldy and not particularly informative.
Pertinent PMHx of all 30 patients |
Mild COPD not on O2 |
None |
None |
None |
Mild Intermittent Asthma |
Myelofibrosis |
Bilateral Lung Transplant |
Obscure GIB |
HFrEF, Anemia |
Heart/Kidney Transplant |
HFpEF, COPD, Hypertension |
Liver Transplant, UTI |
CTD-ILD not on O2 |
HFpEF, PE, Meth Use |
Lung Transplant |
Hypertension, DM2 |
PE |
Lung CA, PE, COPD |
Hypertension, AKI, Crohn's |
Mild Intermittent Asthma |
DM2 |
CKD3b, ESBL UTI |
HFrEF w/ LVAD |
PE, Afib |
DM2, Hypothyroidism |
HFpEF, AS, DM2, Hypertension |
Afib, DM2 |
B-Cell ALL |
EtOH Use, Hypertension, Gout |
DM2, Hypertension |
- Table 1. Was any of the patient taking medications (i.e., vasodilators) which could affect pulse oximetry readings?
No patients were taking vasodilators or vasoconstrictive medications. Additionally, no patients had received methylene blue, indocyanine green, or other dyes that could affect the accuracy of the pulse oximeter. Finally, no patients had hemoglobinopathies (genetic or acquired) that would interfere with pulse oximetry. We have added a section to the methods to clarify this point.
- Table 1. Do you ABG values. Did anybody come with Type 2 respiratory failure? How would OD perform in these patients?
We do not have ABG values in these patients as they were on room air almost exclusively in the ED and not severely ill enough to necessitate ABGs. Moreover, the decision to obtain ABGs was also left to the treating physicians who were blinded to the presence of the study.
This is an excellent question. We do not know if patients had hypercarbic respiratory failure as ABGs were not ordered and patients aren’t routinely placed on end-tidal CO2 monitoring in our hospital. From our prior study (doi:10.1152/ajplung.00371.2018) we do know that the PCO2 as reported by the AGM100 correlates well with the PaCO2 as measured by ABG -3.6mmHg bias across a wide range of PCO2 values. Most importantly, however, the PCO2 is explicitly factored into the OD calculation (see manuscript ref #10) and so shifts in the O2-Hb dissociation curve due to changes in PCO2 are directly dealt with. Thus, irrespective of the PCO2 (high or low), the OD takes it into account and therefore the presence of an abnormal CO2 should not significantly alter the performance of the OD.
- Results. Figures 3 and 5. Could you please compare the AUCs of the SpO2 and OD? Consider DeLong test. There is an overlap in CIs of the AUCs, therefore I feel the curves are not different.
Thank you for this excellent suggestion. Although the curves for ODFlip and SpO2 look similar, they only overlap at extremes of Sn and 1-Sp and at the vertex, or Youden index, they clearly diverge supporting a relevant difference between the two classifiers even if not statistically significant. Having said that, we reanalyzed our data, as requested, in R to perform a DeLong test (SPSS lacks this capacity). We only did this for supplemental oxygen administration as this was the primary outcome of interest (please see comments above) and to expedite our response to reviewers. This analysis demonstrated no significant difference between OD (as displayed by ODFlip) and SpO2 and we have reported these methods and results in the appropriate sections of the manuscript. However, the Z statistic is reasonable, and the p-value is low so this may be an issue of a rather small sample size. A second consideration is that the DeLong method prone to Type-II error, which is supported by this manuscript. (doi: 10.1002/sim.5328).
Reviewer 2 Report
Major
Coronavirus disease 2019 (COVID-19) is a predominantly respiratory infectious disease caused by SARS-COV-2, which mainly affected the respiratory system. The clinical presentation of COVID-19 is highly heterogeneous, ranging from asymptomatic to severe respiratory failure. People with respiratory failure caused by COVID-19 could deteriorate to requiring invasive mechanical ventilation or death.
However, it is not easy to differentiate the severe COVID-19 patients from asymptomatic patients by their symptoms. Compared to typical ARDS, some critically ill COVID-19 patients do not present obvious dyspnea even though they have severe hypoxemia, which is clinically called “happy hypoxemia”.
The assessment of arterial oxygen saturation by Pulse oximetry (SpO2) is a candidate mean for the deterioration type of COVID-19. However, the assessment of SpO2 has not been always effective for clinical decision making for the therapeutic decision. The authors have focused on new index such as alveolar gas monitor (AGM100) adds exhaled gas tensions to SpO2 to calculate the oxygen deficit (OD). The OD parallels the alveolar-to-arterial oxygen difference (AaDO2) in outpatients with cardiopulmonary disease. They comparted the OD between admitted versus discharged patients and between patients who required supplemental oxygen and those who did not. The mean OD was significantly higher among those admitted 50.0±20.6 vs discharged 27.0±14.3. The score changes are almost double. The OD was also significantly higher among those requiring supplemental oxygen 60.1±12.9 vs those remaining on room air 25.2±11.9. The ROC curves for the OD demonstrated very good and excellent sensitivity for predicting hospital admission and supplemental oxygen administration, respectively.
The authors have concluded that the AGM100 is a novel, noninvasive way of measuring impaired gas exchange for clinically important endpoints in COVID19.
Since the A-a Do2 may be a most sensitive method for gas exchange abnormality in respiratory diseases, The AGM may be an effective indicator for decision making for the therapy for COVID-19.
One problem is that symptom severity and ARDS frequency caused by COVID-19 are very different among SARS-Cov-2 variants. Thus, this method may be applicable for a type of COVID-19, which has severely affected the respiratory systems.
The authors should discuss the symptom variability based on the SARS-Cov-2 variants.
Author Response
Coronavirus disease 2019 (COVID-19) is a predominantly respiratory infectious disease caused by SARS-COV-2, which mainly affected the respiratory system. The clinical presentation of COVID-19 is highly heterogeneous, ranging from asymptomatic to severe respiratory failure. People with respiratory failure caused by COVID-19 could deteriorate to requiring invasive mechanical ventilation or death.
However, it is not easy to differentiate the severe COVID-19 patients from asymptomatic patients by their symptoms. Compared to typical ARDS, some critically ill COVID-19 patients do not present obvious dyspnea even though they have severe hypoxemia, which is clinically called “happy hypoxemia”.
The assessment of arterial oxygen saturation by Pulse oximetry (SpO2) is a candidate mean for the deterioration type of COVID-19. However, the assessment of SpO2 has not been always effective for clinical decision making for the therapeutic decision. The authors have focused on new index such as alveolar gas monitor (AGM100) adds exhaled gas tensions to SpO2 to calculate the oxygen deficit (OD). The OD parallels the alveolar-to-arterial oxygen difference (AaDO2) in outpatients with cardiopulmonary disease. They comparted the OD between admitted versus discharged patients and between patients who required supplemental oxygen and those who did not. The mean OD was significantly higher among those admitted 50.0±20.6 vs discharged 27.0±14.3. The score changes are almost double. The OD was also significantly higher among those requiring supplemental oxygen 60.1±12.9 vs those remaining on room air 25.2±11.9. The ROC curves for the OD demonstrated very good and excellent sensitivity for predicting hospital admission and supplemental oxygen administration, respectively.
The authors have concluded that the AGM100 is a novel, noninvasive way of measuring impaired gas exchange for clinically important endpoints in COVID19.
Since the A-a Do2 may be a most sensitive method for gas exchange abnormality in respiratory diseases, The AGM may be an effective indicator for decision making for the therapy for COVID-19.
One problem is that symptom severity and ARDS frequency caused by COVID-19 are very different among SARS-Cov-2 variants. Thus, this method may be applicable for a type of COVID-19, which has severely affected the respiratory systems.
The authors should discuss the symptom variability based on the SARS-Cov-2 variants.
We acknowledge that the delta variant of COVID had higher lethality than other strains. Additionally, we agree that some strains were more likely to cause anosmia or GI symptoms than others. However, our experiment was focused on the early gas exchange abnormalities of COVID predicated on detecting early parenchymal involvement, which is/was possible with each strain. Therefore, we did not record the ultimate strain of each positive COVID test in our cohort. Based on when patients were enrolled, we presume that we captured patients with the alpha, beta, delta, and omicron strains. Ultimately, we feel this technology is applicable to all strains of COVID as well as additional potential respiratory pathogens.
Round 2
Reviewer 1 Report
I am happy with the changes and suggest acceptance.